# Updated Virophage Taxonomy and Distinction from Polinton-like Viruses

**DOI:** 10.3390/biom13020204

**Published:** 2023-01-19

**Authors:** Simon Roux, Matthias G. Fischer, Thomas Hackl, Laura A. Katz, Frederik Schulz, Natalya Yutin

**Affiliations:** 1DOE Joint Genome Institute, Lawrence Berkeley National Laboratory, Berkeley, CA 94720, USA; 2Max Planck Institute for Medical Research, Department of Biomolecular Mechanisms, 69120 Heidelberg, Germany; 3Groningen Institute of Evolutionary Life Sciences, University of Groningen, 9700 AB Groningen, The Netherlands; 4Department of Biological Sciences, Smith College, Northampton, MA 01063, USA; 5National Center for Biotechnology Information, National Library of Medicine, National Institutes of Health, Bethesda, MD 20894, USA

**Keywords:** virophage, polinton, polintovirus, giant virus, virus taxonomy

## Abstract

Virophages are small dsDNA viruses that hijack the machinery of giant viruses during the co-infection of a protist (i.e., microeukaryotic) host and represent an exceptional case of “hyperparasitism” in the viral world. While only a handful of virophages have been isolated, a vast diversity of virophage-like sequences have been uncovered from diverse metagenomes. Their wide ecological distribution, idiosyncratic infection and replication strategy, ability to integrate into protist and giant virus genomes and potential role in antiviral defense have made virophages a topic of broad interest. However, one limitation for further studies is the lack of clarity regarding the nomenclature and taxonomy of this group of viruses. Specifically, virophages have been linked in the literature to other “virophage-like” mobile genetic elements and viruses, including polinton-like viruses (PLVs), but there are no formal demarcation criteria and proper nomenclature for either group, i.e., virophage or PLVs. Here, as part of the ICTV Virophage Study Group, we leverage a large set of genomes gathered from published datasets as well as newly generated protist genomes to propose delineation criteria and classification methods at multiple taxonomic ranks for virophages ‘sensu stricto’, i.e., genomes related to the prototype isolates Sputnik and mavirus. Based on a combination of comparative genomics and phylogenetic analyses, we show that this group of virophages forms a cohesive taxon that we propose to establish at the class level and suggest a subdivision into four orders and seven families with distinctive ecogenomic features. Finally, to illustrate how the proposed delineation criteria and classification method would be used, we apply these to two recently published datasets, which we show include both virophages and other virophage-related elements. Overall, we see this proposed classification as a necessary first step to provide a robust taxonomic framework in this area of the virosphere, which will need to be expanded in the future to cover other virophage-related viruses such as PLVs.

## 1. Introduction

“Virophage” is a generic term currently used to describe viruses with dsDNA genomes that are able to co-infect a eukaryotic cell alongside a giant virus and then hijack parts of the giant virus machinery for their own replication [1,2,3]. The first virophage was discovered in an amoeba host (*Acanthamoeba castellanii*) alongside the giant Mamavirus and was named Sputnik because of its satellite-like reliance on the Mamavirus machinery [4]. A second virophage, named mavirus, was characterized soon after [5]. It was found co-infecting the marine flagellate *Cafeteria* sp. with the giant Cafeteria roenbergensis virus (CroV) and revealed an evolutionary link between virophages and eukaryotic mobile genetic elements of the *Maverick*/*Polinton* class. A handful of additional virophages have been cultivated and characterized since, including multiple Sputnik-like and Zamilon virophages associated with different mimivirus strains, and more recently, the first virophage of a green alga, *Chlorella* virus virophage SW01 (CVv-SW01), which depends on the CroV-like Chlorella Virus XW01 [2,6,7]. For clarity, the term “virophage” will be used hereafter to designate specifically viruses related to Sputnik, Zamilon, mavirus, and CVv-SW01.

Comparative analysis of isolated virophages outlined several genomic and phenotypic defining characteristics. Their double-stranded (ds) DNA genomes are around 20 kilobases (kb) long, feature a low GC content, and encode for ≈20 predicted proteins [2,8]. Their major capsid protein adopts a double jelly-roll (DJR) fold and forms the shell of a 50 to 75 nm-wide icosahedral virion. In addition to recognizable structural genes (e.g., major and minor capsid proteins), all virophages encode two other proteins likely involved in encapsidation, an Adenain-like maturation protease, and an FtsK-HerA-like ATPase. Finally, these virophages only replicate when their protist host is co-infected with a giant virus, specifically from the *Imitervirales* order. The extent of the negative effect of virophage replication on the giant virus replication and burst size varies, as co-infection results in severe inhibition of the giant virus in the case of Sputnik and mavirus, but not for Zamilon [3]. These defining characteristics were the basis for the initial taxonomic classification of virophages, proposed in 2016 [9]. Briefly, one family (*Lavidaviridae*) and two genera (*Sputnikvirus* and *Mavirus*) were established to include the three recognized dsDNA virophage species at the time: *Cafeteriavirus-dependent mavirus*, *Mimivirus-dependent virus Sputnik*, and *Mimivirus-dependent virus Zamilon*. This taxonomy was amended in 2020 with the addition of higher ranks that connected existing virophage taxa to the larger *Preplasmiviricota* phylum, *Bamfordvirae* kingdom, and *Varidnaviria* realm [10].

The characterization of Sputnik and mavirus also enabled large-scale searches for virophage-like sequences in metagenomes, which uncovered a broad diversity of virophages existing in the environment. Virophage-like sequences were detected across many types of biomes, including human and other animal gut microbiomes [11,12], as well as freshwater lakes from Antarctica [13], North America [14,15,16], Asia [17,18], and Europe [19]. Further comparative genomic studies, including a large compendium of genomes assembled from metagenomes, indicated that these virophage (-like) sequences were diverse and part of a larger evolutionary network including other eukaryotic viruses & mobile genetic elements, most notably polinton-like viruses (PLVs) [19,20,21,22,23]. Like virophages, PLVs seem to be evolutionary related to mobile genetic elements of the *Maverick*/*Polinton* class, can co-infect eukaryotic host cells alongside a giant virus, and seemingly encode a comparable encapsidation machinery although with no detectable sequence similarity to the one of virophages [24,25,26]. Meanwhile, the integration of mavirus-like virophages into their eukaryotic host genomes and subsequent reactivation was further explored and revealed a high degree of mobility, with a potential role as a population-based defense system against their cognate giant viruses [27,28]. The unexpected plethora of virophage-like elements revealed by these different studies resulted in a confusing nomenclature where different labels were given for entities that had similar or overlapping properties. Specifically, virophage-related sequences were alternatively reported as “virophages,” “polintons,” “polintonviruses,” “polinton-like viruses,” “adintoviruses,” or “MELD viruses” [19,23,24,29]. The use of different names reflects the current lack of a taxonomic framework for virophages and related elements, owing to the fast increase of metagenomic sequence reports in recent years.

Here, in support of the ICTV virophage study group, we reanalyze a comprehensive set of virophage-like metagenome-derived genomes complemented with additional host-associated datasets to provide a unified identification and classification framework. Specifically, we first update the virophage taxon demarcation criteria to be applicable to uncultivated virophages, then we evaluate how phylogenetic and gene content analyses can be combined to identify robust clades within this virophage taxon, and finally, we explore how these metagenome-derived sequences, taken together, inform on the ecological distribution and host range of virophages.

## 2. Results & Discussion

### 2.1. Definition of a Formal Virophage Taxon and Associated Demarcation Criteria

To be operationally useful for metagenome-derived sequences, taxon delineation criteria should be based on genome features only and complemented by additional information if/when available. The current official demarcation criteria of the *Lavidaviridae* family, to which all virophages belong, include both genetic markers and phenotypic information (a virus that is “dependent on or associated with, a large dsDNA virus related to the so-called nucleo-cytoplasmic large DNA viruses”) [9,10]. The genetic markers criteria highlighted four genes from the morphogenetic module that were shared by all known virophages and can be used in phylogenetic analysis to demonstrate genetic similarity and common origin. These four genes include virophage-specific versions of (i) major capsid protein (MCP), also known as “hexon” protein, (ii) minor capsid protein (minCP), also known as “penton” protein, (iii) FtsK-HerA family DNA-packaging ATPase (ATPase), and (iv) maturation cysteine protease (PRO), also known as Adenain. Since these demarcation criteria were first proposed, several studies confirmed the co-occurrence of MCP, ATPase, and PRO in complete or near-complete genomes [6,12]. The virophage penton protein is also most often detected, with some rare exceptions where it seems to have diverged beyond recognition, as in the rumen virophages (RMV) [11].

To confirm whether gene content alone could be used to distinguish virophages such as Sputnik and mavirus from other related elements, we first searched the 36 sequences currently assigned to the *Lavidaviridae* family in GenBank and the largest RMV sequence using HMM profiles previously established for each of these marker genes [12]. Most lavidavirus genomes included the four virophage marker genes, with the known exception of RMVs lacking an identifiable virophage-like penton protein (Figure 1). Four genomes encoded none of these markers, namely Phaeocystis globosa virus virophage and three Chrysochromulina parva virus virophages. These genomes were already known to be more similar to polinton-like viruses and thus would logically not belong to the same taxon as other virophages [11]; however, they provide a good opportunity to formally define the new taxon boundaries. Specifically, no gene of these polinton-like viruses showed significant similarity to any of the virophage marker genes (highest hmmsearch scores < 10), while virophage sequences typically displayed hmmsearch scores ≥ 100, except for the penton protein (Figure 1B). We thus propose to update the delineation criteria for a virophage taxon as follows:–required features: the complete genome should encode a virophage-like hexon protein, a virophage-like ATPase, and a virophage-like cysteine protease, all of which can be detected based on established HMM profiles for each of these marker genes–other expected (but not required) features: the genome should consist of dsDNA with a length between 15 kb and 45 kb and encode a virophage-like penton protein detected based on established HMM profile(s).

### 2.2. Common Origin and Genetic Diversity in the Extent Virophage Clade

To explore the genetic diversity within the virophage clade as defined by the demarcation criteria above, we used the same HMM profiles of the four virophage marker genes to gather virophage sequences from previously published datasets [12,28,30], public databases (as of October 2021), and newly-sequenced protist genomes (Appendix A). Specifically, we retained all sequences for which at least three of the four virophage marker genes could be detected. The resulting 1869 sequences were clustered into 848 distinct viral operational taxonomic units (vOTUs) using standard cutoffs for dsDNA viruses, i.e., 95% average nucleotide identity (ANI) and 85% alignment fraction (AF) [31]. Among these, 257 vOTU representatives were predicted to be complete or near-complete (Appendix A, and see Section 4. Methods). All 257 (near-)complete genomes encoded the MCP and ATPase genes, 256 encoded a PRO gene, and 247 also encoded a recognizable penton protein (minCP), confirming that virophages typically encode all 4 of these marker genes.

We first used this dataset in phylogenetic analyses to confirm that virophages form a cohesive taxon (Appendix A). In trees built from the ATPase and PRO genes, the two markers for which alignable homologs can be identified in other viruses and/or cellular genomes, all virophage sequences formed a monophyletic group (Appendix A). The only exception was an ATPase encoded on a contig from a Pithovirus MAG (LCPAC001) that branched within the virophage clade and may be a genuine virophage erroneously binned in a giant virus genome (see below). Overall, these two rooted trees confirmed that virophages could be classified as a single monophyletic taxon and reinforced the use of the four marker genes for virophage classification.

Next, we performed de novo protein clustering to estimate the diversity of genes encoded in virophage genomes. This clustering highlighted the four marker genes from the morphogenesis gene module as the only near-universally conserved ones in the dataset. No other genes were identified in more than half of the genomes, even when using sensitive clustering approaches (Figure 2A). Specifically, each (near-)complete virophage genome encodes, on average, 27 predicted proteins with a majority either unique to a single vOTU (n = 6) or detected in less than 10% of virophage vOTUs (n = 8), highlighting the high level of divergence between virophage genomes. A similar pattern can be observed when analyzing pairwise similarity for the morphogenesis genes: while each gene can readily be identified via key conserved residues, most virophage pairs show less than 30% amino-acid sequence similarity in their MCP, minCP, PRO, or ATPase (Figure 2B). Taken together, the low similarity in primary sequence, even for conserved genes, coupled with the high diversity in other genes indicates that, despite their likely common origin and recognizable set of conserved genes, virophage genomes encode a large and mostly idiosyncratic genetic diversity. We propose that this level of diversity justifies the establishment of a taxon at the class level, i.e., *Maveriviricetes*. The demarcation criteria listed above should thus be interpreted as defining whether a virus belongs to the *Maveriviricetes* class.

### 2.3. Challenges and Limitations of Taxonomic Classification within the Virophage Clade

As illustrated above by the diverse gene content and low level of similarity even for conserved genes, establishing robust lower-level taxa within the virophage clade, i.e., within the *Maveriviricetes* class, is challenging. The current ICTV taxonomy defines one genus each for *Sputnik* and *Mavirus* virophage species, which is not sufficient to handle the additional diversity described. Here, we explored whether consistent clades can be derived from a phylogeny based on MCP, the most conserved gene across virophages, and gene content clustering based on genome-wide amino acid identity (gwAAI). This analysis was restricted to the set of 257 virophage genomes identified as complete or near-complete (see above).

The major capsid protein (MCP) was selected for phylogeny reconstruction as it was the longest of the four genes in the morphogenesis gene module (Appendix A). Consistent with previous studies, the MCP phylogeny was difficult to resolve due to high sequence divergence, and the resulting tree included some long branches with uncertain placement. Nevertheless, for most of the tree, robust monophyletic clades with strong (>90) bootstrap support can be delineated and used to form the basis of formal taxa (Figure 3A). To identify which clades were the most appropriate to define as taxa, we added complementary information, including genome features (genome length, MCP length, presence of a recognizable penton), clustering based on gwAAI, and source environment, to the MCP tree, with the rationale that any taxon should ideally have distinguishing characteristics beyond grouping in a monophyletic clade in the tree (Figure 3A).

Combining phylogeny and genome metadata enabled us to delineate seven distinct groups within this dataset (Figure 3A,B). The first one noted as “mavirus virophage” in Figure 3B, represents virophages branching next to the mavirus isolate and clustered in the same group (“gwAAI Group_006”) based on gwAAI. It includes 11 genomes, is only distantly related to other virophages on the MCP tree, and is mostly composed of sequences integrated into *Cafeteria* genomes. Similarly, a small group noted as “Rumen virophage” is composed of 6 genomes clustered in the same gwAAI group (“gwAAI Group_007”) and only distantly related to other virophages on the MCP tree. This second group includes the sequences originally described as “rumen virophages” [11], along with other metagenome-derived, genomes mostly from rumen samples (Figure 3A and Appendix A). The third group (“Sputnik virophages”) is formed by a monophyletic clade of 47 genomes, including Zamilon and Sputnik virophages, the majority (all but 2) clustered in “AAI Group_002”. This group also displays short genomes (median = 18.9 kb) and a relatively low GC content (median = 31.3%) compared to other virophages (Appendix A). The fourth group noted as “large virophage,” forms a distinct clade with characteristically large genomes (median = 31.4 kb) and MCPs (median length = 938 aa), as well as unusually high GC content (median = 55.2%, Figure 3A and Appendix A). Finally, three groups noted as “SW01 virophages”, “aquatic virophages 1”, and “aquatic virophages 2,” include 146 genomes branching next to the “large virophages” but displaying genome lengths, MCP lengths, and GC content comparable to other virophages, and clustered into three distinct gwAAI groups (Figure 3A,B and Appendix A). The former (“SW01 virophages”) includes the recently isolated Chlorella virophage SW01, while the other two lack an isolated representative [6].

In addition to forming monophyletic clades on an MCP tree, we further verified whether members of these groups also formed monophyletic clades on trees built from the three other virophage morphogenesis genes. The divergence level for these markers led to sub-optimal alignments and consistently long branches so that the deeper structure of the trees remained uncertain. Nevertheless, all clades remain broadly monophyletic on the ATPase and penton trees, with only a handful of inconsistent sequences (Appendix A). In the PRO tree, rumen virophages branched within SW01 virophages, and mavirus virophages branched within Sputnik virophages, but both rumen and mavirus virophages were associated with long branches and lower node support (often < 80). This suggests that, while clades overall remained consistent across markers, their relative position to one another is challenging to resolve, given the level of diversity in the different alignments. This is also true for the MCP tree, where some genomes display long branches that cannot be placed with certainty (e.g., Yellowstone Lake virophage 7).

Given the robustness of the seven delineated clades, and as part of the ICTV virophage sub-group, we will propose to establish these as formal taxa at the order and family level, as follows:–Rumen virophages as the new *Ruviroviridae* family in the new *Divpevirales* order, named for virophages with Divergent penton proteins.–Mavirus virophages as the new *Maviroviridae* family after the first isolated member of the taxon, in the new *Lavidavirales* order, a name adapted from the current *Lavidaviridae* family named for “Large virus dependent or associated.”–Sputnik virophages as the new *Sputniviroviridae* family after the first isolated member of the taxon, in the new *Mividavirales* order, for “Mimivirus dependent or associated”–SW01 virophages as the new *Dishuiviroviridae* family after the lake from which the first member of the taxon (SW01) was isolated, in the *Priklausovirales* order, which is currently the only order established in the *Maveriviricetes* class.–Aquatic virophages 1 as the new *Omnilimnoviroviridae* family using the prefixes “omni” (“all,” “everywhere”) and “limno” denoting a link to freshwater environments, since members of this clade were detected across a broad geographic range of freshwater lakes, also in the same *Priklausovirales* order.–Large virophages as the new *Gulliviroviridae* family named after Lemuel Gulliver, the main character of “Gulliver’s travel,” since these virophages can be considered as “giant” compared to other virophages but are still relatively small compared to their associated giant viruses, also in the same *Priklausovirales* order.–Aquatic virophages 2 as the new *Burtonviroviridae* family named after Mary Burton, the wife of Lemuel Gulliver in “Gulliver’s travel,” as this clade is the most closely related to the large virophages, also in the same *Priklausovirales* order.

The decision to group the latter four clades in a single order (*Priklausovirales*), and separate all others, was based on the divergence between clades observed in the tree and the pairwise MCP AAI (Appendix A).

### 2.4. Examples of Detection and Taxonomic Assignment of Virophages in Mixed Datasets

Building on this new framework, the next step was to establish criteria and operational processes to assign new sequences to these virophage taxa. Ideally, new virophage sequences would be classified based on similar approaches as the ones used above to delineate taxa, i.e., combining multiple phylogenies, genome features, and genome-wide AAI clustering. We recognize, however, that this approach may be impractical in some cases, and an alternative method to assign new sequences to existing virophage taxa based on a simple sequence comparison to reference profiles or databases would be valuable. Based on the new proposed taxa and demarcation criteria outlined above, we built and benchmarked such an approach and provided it as an integrated package including relevant databases and an automated classifier (Figure 4A, https://github.com/simroux/ICTV_VirophageSG, accessed on 15 November 2022).

To determine whether a new sequence belongs to the *Maveriviricetes* class as defined above, we built updated HMM profiles for each of the four morphogenesis genes and specifically recommend using the detection of the MCP profile as the main demarcation criteria, with the three other markers primarily used to confirm this assignment and providing a measure of genome completeness. Within *Maveriviricetes*, we propose assigning new sequences to the seven taxa using a best BlastP hit approach to MCP references, with cutoffs derived from comparing complete and near-complete genomes (Figure 4A, Appendix A). This “best hit” approach was selected rather than a clustering based on MCP pairwise AAI due to the limited ability of the latter to distinguish “within taxa” from “between taxa” comparisons (Appendix A). When applied to the set of reference sequences and omitting self-hits, BLAST-based affiliations recapitulated the classification of all original members of each family defined based on the MCP phylogeny (Figure 3B). Hence, while not strictly identical, this BLAST-based assignment should be a useful approximation to the phylogeny-driven families defined above. Finally, to help further distinguish between virophages within the *Maveriviricetes* class and Polinton-like viruses (PLVs), we included in the classifier a comparison to a previously published HMM profile corresponding to an FtsK-HerA ATPase conserved across PLVs [12,26]. Sequences not assigned to *Maveriviricetes* can thus be flagged as possible PLVs based on the detection of this profile (Figure 4A).

To illustrate how the process would work in practice, we first applied this BLAST-based assignment to the partial sequences in our dataset, i.e., virophage sequences that were not identified as complete or near-complete and thus not included in the main MCP phylogeny. Overall, 393 (66%) of these sequences could be assigned at the family level, with the majority classified in the new Burtonviroviridae proposed family (31% of the assigned sequences, Appendix A). As further confirmation, we performed a phylogeny-based affiliation for these same sequences, which was nearly identical to the BLAST-based approach (92% of sequences with identical assignment, Appendix A). We next applied the BLAST-based assignment approach to a dataset obtained from a high-altitude mountain lake in which virophage-like sequences had been previously identified via a custom gene content approach [19]. Overall, 31 sequences were classified in the *Maveriviricetes* class based on the detection of a virophage MCP, including 26 for which all four marker genes could be detected and another three for which three of the four marker genes could be detected (Figure 4A, Appendix A). These 31 sequences corresponded almost exactly to the list of 32 virophages highlighted in the original study, with the only one sequence missing being a partial genome in which no MCP was detected. We also applied the same approach to previously published “Adintoviruses” (n = 64), of which several were classified as *Polintoviricetes*, another class next to the *Maveriviricetes* within the *Preplasmiviricota* phylum [23,32,33]. Among these, five sequences were classified as *Maveriviricetes* with our approach, all displaying three or four of the virophage morphogenesis genes. Genome comparison further confirmed that four of these sequences were similar to rumen virophages (n = 4), which would justify a reclassification in the *Maveriviricetes* taxon (Appendix A).

For the detection of PLVs, 53 of the 77 sequences that had been previously identified as “PLV” in the Gossenköllesee dataset displayed a hit to the PLV FtsK-HerA ATPase profile, while none of the virophages did (Figure 4B,C). Similarly, 39 of the 64 adintoviruses displayed a hit to this PLV profile, including the two ICTV-recognized exemplar genomes for the *Polintoviricetes* class: Mayetiola barley midge adintovirus strain 2012 and Terrapene box turtle adintovirus isolate 5272. This suggests that the *Polintoviricetes* class may eventually serve as a formal taxon for most, if not all, PLVs. Notably, PgVV and the three Chrysochromulina parva virus virophages, which are currently classified in the *Lavidaviridae* in GenBank, do not fulfill the criteria described above for classification into the *Maveriviricetes* (Figure 1) and display a significant hit to this PLV profile, suggesting they should be reclassified into the *Polintoviricetes* class.

Finally, while identifying virophages in public databases, we noticed 12 contigs fulfilling all criteria for classification in *Maveriviricetes* but currently classified as Bacteria or Archaea (Appendix A—abbreviated sequence ids starting with “MAG_”). All these contigs originated from metagenome-assembled genomes (MAGs) and likely reflected erroneous binning. This indicates that virophage contigs represent an underestimated source of prokaryotic MAG contamination, which may stay undetected by current tools.

### 2.5. Beyond Genome-Based Taxonomy: Virophage Host Range and Interactions

A better characterization of virophage and virophage-like evolutionary history and taxonomy is especially important to better understand the impact and long-term association of these elements with their giant virus and eukaryotic hosts. Both Sputnik and mavirus hijack the molecular machinery of their associated giant viruses and, in so doing, likely reduce the burden imposed on the protist host community by these eukaryotic viruses. A recent PLV-like element showed a similar association with both a giant virus and a eukaryotic virus and a reduction in the fitness of the giant virus, suggesting this lifestyle and infection cycle may be a common feature of virophages and PLVs [25]. However, most virophages are only known from metagenome assemblies; thus we lack information about their potential giant virus and eukaryotic hosts. Mining eukaryotic genomes, either from isolates or obtained through single-cell genomics, can help bridge this gap.

Our virophage dataset included 36 sequences identified in eukaryotic genomes obtained from cultivated strains. A majority (34 of 36) were detected in *Cafeteria burkhardae* (formerly *C. roenbergensis*) genomes, including 11 integrated into a larger host region, as previously reported [28]. Notably, three metagenome-derived virophage genomes were similarly identified on larger contigs, including a likely host region of 5 to 10 kb (Appendix A). While these host regions are similar to various protist taxa and do not allow a clear host prediction for these virophages, they suggest that additional instances of virophage integration could be discovered from metagenomes in the near future. The two other virophages detected in eukaryotic genomes were detected in the genomes of *Caecitellus* and an *Adriamonas*-like bicosoecid, marine, and freshwater heterotrophic flagellates, respectively (Appendix A; Hackl, Arndt, Fischer, unpublished). According to the BLAST-based approach (see above), the former was classified as an aquatic virophage group 2 (proposed Burtonviroviridae), while the latter remained unclassified within the *Maveriviricetes* class.

Since microeukaryotes are difficult to cultivate, however, single-cell genomics provides an alternative option to obtain genome sequences for individual microeukaryote cells either automatically sorted or handpicked from environmental samples. Our virophage dataset included 32 sequences obtained from partially sequenced single-cell genomes, providing additional indications of the type of eukaryotic hosts found across virophage diversity (Appendix A; Katz Lab, unpublished). These 32 sequences originated from eight genomes of the testate amoeba *Hyalosphenia elegans* (n = 16), six genomes from *Loxodes* spp. (n = 13), and two genomes from *Halteria* spp. (n = 3). Virophages classified in the proposed Sputniviroviridae, Dishuiviroviridae, Omnilimnoviroviridae, and Burtonviroviridae families were identified, with some single-cell genomes including multiple distinct virophages suggesting the potential for co-infection. While these data point to additional hosts of virophages, such associations would have to be confirmed experimentally, as virophages could also be detected in single-cell genomes if they were attached to or engulfed in but not infecting the protist cell. Building on these candidate associations, further characterization of virophage-host interaction would ideally be performed using recently developed in vitro assays such as virusFISH [34] and droplet digital PCR [35].

## 3. Conclusions

Virophages are part of a fascinating and complex network of smaller dsDNA viruses that are associated with giant viruses and eukaryotic hosts. Based on metagenome analysis, virophages are a diverse group of viruses with a complex evolutionary history, which led to confusion regarding the extent, nomenclature, and characteristics of virophages as a taxonomic clade. As members of the ICTV virophage study group, we focused here on sequences similar to the virophages stricto sensu and sought to establish updated demarcation criteria that can be applied to both cultivated and uncultivated virophages. We then attempted to delineate robust taxa within this virophage clade based on the recently published datasets of metagenome-derived virophage genomes. This led us to propose that virophages, i.e., viruses similar to Sputnik and mavirus, should be placed in the *Maveriviricetes* clade and delineated based primarily on the double jelly-roll major capsid protein version that is unique to this clade, and on three other morphogenesis genes (minCP, ATPase, PRO) typically also encoded on these genomes. Within this class, we further propose to distinguish seven family-level groups, to be soon formally proposed to ICTV, delineated based on a combination of phylogenetic and genome clustering approaches. Finally, we provide sequence-similarity-based approaches to assign newly sequenced virophages to these different groups. A similar framework could likely be used for PLVs, some of them currently classified in the *Polintoviricetes*, which would further clarify taxonomic classification in this area of the virosphere. There is much that remains to be discovered and characterized in terms of virophage and PLV biology, including host range, interaction with giant viruses and eukaryotic hosts, or potential new defense/counter-defense systems [25,36]. Ultimately, a comprehensive genome catalog of virophages, along with a clear and updated taxonomic scheme, represents a framework that will facilitate future studies and more in-depth characterization of these fascinating viruses.

## 4. Methods

### 4.1. Collection of Virophage and Virophage-Like Sequences

Virophage and virophage-like genomes used in this study (Appendix A) were collected from multiple sources. First, all genomes classified as *Lavidaviridae* in the NCBI Nucleotide database were downloaded in March 2022, excluding partial genomes MN151334 and KJ183141, which only include a portion of the major capsid protein. This *Lavidaviridae* genome reference dataset (n = 28) also included sequences downloaded in 2014 from NCBI Nucleotide, namely Yellowstone Lake virophages 1 to 4 and Ace Lake virophage. Next, all sequences assigned as *Lavidaviridae* in IMG/VR v3 [30] were collected (n = 1417), as well as a large set of metagenome-derived virophage sequences published in 2019 and partially overlapping with IMG/VR v3 [12] (n = 331), and a set of unique virophage sequences described in 2015 from rumen samples [11] (n = 7). Finally, we included a set of recently described integrated virophages from *Cafeteria burkhardae* [28] (“EMALEs,” n = 6). All these sequences were verified to encode at least three of the four members of the virophage morphogenesis module, previously highlighted as identification criteria for members of the *Lavidaviridae* family [9]. This identification was based on published HMM profiles [12], searched with hmmsearch v3.3.2 [37] using a maximum E-value of 0.01 and a minimum score of 40. For all metagenome sequences, ecosystem information was derived from the Gold database [38] or the annotation information available in NCBI RefSeq [39].

To complement these published virophages, we added two sets of newly detected virophage sequences. First, we used published virophage major capsid protein (MCP) sequences in a BlastP search against the NCBI nr database (September 2021), identifying 96 putative virophages. After removing sequences identical to references previously collected and sequences not encoding at least three of the four conserved virophage genes (see above), a total of 46 additional virophage sequences were collected, including 28 detected in whole-genome sequencing of *Cafeteria burkhardae* and overlapping with the “EMALEs” dataset [28], 13 belonging to bacterial and archaeal MAGs and likely resulting from erroneous genome binning, and five metagenome-derived virus genomes currently classified in the *Adintoviridae* family or the MELD group [23,32,33]. Finally, we also included in our dataset 34 virophage sequences, i.e., including at least three of the four conserved virophage genes identified in newly sequenced genomes from cultivated protists (n = 2) or single-cell genomes obtained by whole genome amplification from hand-picked amoeba and ciliates (n = 32). These virophage sequences were included as they may be uniquely associated with a potential host, which could eventually be used as part of a taxonomic classification. More information about these protist genomes is included in Appendix A.

### 4.2. External Datasets Used to Illustrate Virophage Identification and Classification Approach

Two published datasets were used to illustrate the identification and classification approach proposed here because they had been presented as including virophage and other related viruses [19] or noticed to do so based on our search of NCBI nr database [23,32,33]. For the former dataset, identified as “Gossenköllesee,” all new contigs identified by the authors as virophages or PLV were downloaded from https://figshare.com/s/9a7a1d16d77ea9d658a1, accessed on 23 September 2022, and processed through the identification and classification pipeline (n = 114). The original affiliation of these contigs was obtained from Appendix A in [19]. For the latter datasets, all contigs classified in the *Adintoviridae* family and available in NCBI GenBank in August 2022 were collected (n = 28) as well as the contigs listed in [23], which include additional unclassified viruses (n = 59). The two datasets were combined and dereplicated using MUMmer v4.0.0b2 [40] at 99% average nucleotide identity (ANI) over 99% of the shorter sequence length to obtain a non-redundant dataset of 64 sequences, referred to as “Adintovirus.”

### 4.3. Virophage Genome Clustering, Quality Control, and Trimming

Sequences possibly corresponding to integrated virophages were first refined to trim any possible host region that may be found on the same contig. For all virophages detected in *Cafeteria burkhardae* genomes, a procedure similar to that of [28] was used. Briefly, for each virophage MCP, a region including 50 kb upstream and downstream was extracted. Predicted coding sequences (cds) in this region were compared to cds from the Mavirus reference genomes using BlastP 2.10.0+ [41] with a minimum bit score of 30. A custom script was used to calculate GC content on the same regions with 100 nucleotide sliding windows. Information was combined and individually inspected for each region to define the most likely boundaries of the integrated virophage, i.e., including known virophage genes and corresponding to a clear shift in GC content [28]. For metagenome sequences obtained from the IMG/VR database or the large previously published dataset [12], putative integrated virophages were identified as follows. First, all these contigs were screened for intergenic regions ≥ 1 kb, a common occurrence when annotating eukaryotic genomes with prokaryotic gene predictors (prodigal v2.6.3, “meta” option, see below). The 49 contigs with an intergenic region ≥ 1 kb were visually inspected on the IMG website to identify potential host regions. This led to the trimming of a potential host region for nine contigs (IMG_VR_0446, IMG_VR_0447, IMG_VR_0497, IMG_VR_0676, IMG_VR_0763, PE_122, PE_123, and PE_131, Appendix A). Second, contigs larger than 30 kb were searched for shifts in GC content, as observed in *Cafeteria burkhardae* genomes. This was observed for one contig (IMG_VR_0457), for which the host region was manually identified and trimmed (Appendix A). For these metagenome-derived integrated virophages, cds predicted in the host region were compared to NCBI nr restricted to the Eukaryota taxon using online NCBI BLAST (August 2022) to determine a potential taxonomy for these host regions.

For contigs not identified as integrated proviruses, direct terminal repeats were detected based on identical sequences of at least ten nucleotides in 5′ and 3′ (n = 81). For all these contigs, all the duplicated sequence in 3′ was trimmed to only retain a single genome unit. When needed, the start of the trimmed contig was shifted to avoid cds prediction spanning across the ends of the contig, leading to incomplete protein sequences.

All post-trimming virophage sequences (n = 1869) were next clustered into vOTUs following standard cutoff for dsDNA virus genomes, i.e., 95% ANI over 85% of the shortest sequence [31], using MUMmer v4.0.0b2 [40] to identify sequence similarity. The longest sequence was picked as the representative for each vOTU, except for ones including isolate and/or reference sequences, for which the oldest and/or highest quality reference genome was manually picked as the representative. The final dataset of 848 vOTU representatives, i.e., non-redundant virophage genomes, was used as input for subsequent analyses, including genome annotation, protein clustering, and phylogenies.

### 4.4. Genome Annotation and De Novo Protein Clustering

For all trimmed representative sequences (see above), cds were predicted using prodigal v2.6.3 with the “meta” option [42]. For functional annotation, predicted protein sequences were then compared to the Pdb70 [43], Pfam [44], and SCOPe [45] databases using hhblits 3.1.0 and the following options: -Z 250 -z 1 -b 1 -B 250, minimum probability score of 95 [46,47]. Predicted protein sequences were also clustered de novo with a similar pipeline as previously used on virophages [12]. Briefly, predicted protein sequences were first compared to each other in an all-vs.-all BlastP [41] (v2.10.0+, default parameters). InfoMap [48] (v0.18.25, --two-level option) was then used to obtain a first level of protein clustering (“Protein clusters”). These protein clusters were further compared to each other using hhsearch as follows: first, all members of a cluster were aligned with muscle v3.8.1551 [49] (default parameters), then a profile was built from this alignment using hhmake v3.1.0 [47] with the options -M 50 and -add_cons. Profiles were then compared all-vs-all with hhsearch v3.1.0 [47] with option -norealign and maximum E-value of 0.001. Hits between clusters displaying a probability score ≥ 90 and covering ≥ 50% of the shorter alignment, or a probability score ≥ 99 and covering ≥ 20% of the shorter alignment, were compiled and used as input for a greedy clustering to establish “superclusters,” i.e., groups of related protein clusters.

Individual predicted functional protein annotations were then summarized at the protein cluster and supercluster levels by assigning the cluster/superclusters to the majority assignment among its members if it included more than 33% of its members. Superclusters corresponding to the canonical virophage morphogenesis gene module were identified based on an assignment to 3J26_N (Penton), 3J26_I (MCP), 4EKF_A (PRO), or the Pfam Clan: P-loop_NTPase (CL0023, ATPase). These 4 superclusters were further validated by verifying that all corresponding genes from reference virophage genomes belonged to the expected supercluster, and no other supercluster could be assigned to one of these four domains/clans. These four superclusters were further refined to remove clusters that correspond to partial and/or duplicated genes. This resulted in a final set of three clusters for MCP, one for PRO, two for ATPase, and 11 for Penton.

To obtain a final systematic detection of these core genes across all genomes and for external datasets, HMM profiles were built for these sequences as follows. For each supercluster, protein sequences were compared to each other in an all-vs-all BlastP [41] (v2.10.0+, default parameters), clustered using MCL 14-137 (inflation parameter = 5) [50], and a multiple alignments and HMM profile were built using MAFFT v7.490 [51] and hmmbuild v3.3.2 [37], respectively, for each group of ten sequences or more. This process yielded 19 HMM profiles (7 MCP, 2 PRO, 4 ATPase, 7 Penton), which were used to identify the four morphogenesis genes in each of the 848 non-redundant virophage genomes (hmmsearch v3.3.2 [37], minimum score of 50, best hit by genome). The sequence divergence for each of the four morphogenesis genes was then evaluated by calculating all-vs-all pairwise amino-acid identity percentage with SDT [52] (Feb 2014 version, default parameters).

### 4.5. Rooted Phylogenetic Trees for ATPase and PRO Genes

The ATPase and PRO genes have homologs in non-virophage genomes, which provides an opportunity to verify that virophages form a monophyletic clade with respect to these other sequences. To that end, two categories of outgroup sequences were included: (i) sequences previously used in similar analyses [20], and (ii) NCBI nr sequences not affiliated to virophages (*Lavidaviridae*) but similar to a virophage (BLAST score ≥ 100). For each marker (ATPase and PRO), the resulting virophage and outgroup sequences were clustered using MCL 14-137 (inflation parameter = 4 for PRO and 9 for ATPase) [50], and one representative of each cluster was selected, prioritizing isolate over metagenome-derived genomes.

For each set of representatives, multiple alignments were computed with MAFFT v7.490 [51] (--auto option). The alignment was visually inspected to remove partial and non-homologous sequences (i.e., lacking conserved regions or residues). IQ-Tree v1.5.5 [53] was then used to build a tree for each curated alignment, with automatic detection of the most appropriate substitution matrix and 1000 replicates of ultra-fast bootstraps. The best-fit model was LG+F+R7 for both PRO and ATPase.

### 4.6. Identification and Analysis of Complete and Near-Complete Virophage Genomes

Among the 848 non-redundant virophage genomes, several categories of sequences were considered as likely representing complete and near-complete genomes: (i) reference genomes from isolates (n = 4), (ii) sequences identified as integrated with upstream and downstream host regions ≥ 2 kb (n = 7), (iii) sequences with direct or inverted terminal repeats (n = 118 and n = 8, respectively), (iv) sequences predicted to be ≥90% complete based on CheckV (AAI-based prediction, n = 59), and (v) linear contigs ≥ 25 kb (n = 61). This latter category was based on the median length of predicted complete and near-complete genomes from all other categories (25,168 bp). Overall, 257 sequences were considered complete or near-complete virophage genomes.

These complete and near-complete genomes were used as input for phylogenetic trees and genome-wide clustering to establish groups and potential taxa within the virophages. For phylogenetic trees, the sequences of the four morphogenesis genes detected in the 257 complete and near-complete genomes using the new HMM profiles (see above) were used after excluding all sequences that covered <60% of the HMM profile to remove partial gene predictions. Multiple alignments were then built for each gene using an iterative clustering-alignment-phylogeny procedure specifically adapted for aligning highly diverging sequences [54]. The alignments were then automatically trimmed using clipkit v1.3.0 [55] using the kpi-smart-gap mode to remove uninformative positions, and the trimmed alignments were used as input for tree building with IQ-Tree v2.2.0.3 [56] with automatic detection of the most appropriate substitution matrix, and 1000 replicates of ultra-fast bootstraps. The best-fit model was Q.pfam+F+R7 for PRO, Q.yeast+F+R8 for ATPase, and Q.pfam+F+R8 for both MCP and penton. For the larger MCP phylogeny, including both complete and partial virophage genomes (Appendix A), multiple alignments were computed with MAFFT v7.490 based on the curated multiple alignment including MCP from complete and near-complete genomes only (options “–add” and “–keeplength”) [51], and the phylogeny was built with tree IQ-Tree v2.2.0.3 [56] with similar parameters as described above.

Genome-wide amino acid identity (AAI) clustering was performed as in [57]. Briefly, predicted protein sequences from the 257 complete and near-complete virophages were compared all-vs-all using diamond v0.9.24.125 [58] and the following options: “--evalue 1e-5 --max-target-seqs 10,000–query-cover 50–subject-cover 50”. The resulting file was used as input for the script “amino_acid_identity.py” to calculate the average AAI for all pairs of genomes. The script “filter_aai.py” was then used to select only pairs of genomes with a minimum normalized cumulative bit score of 0.05. Finally, these selected pairwise AAI values were used as input for an MCL clustering using MCL 14-137 (inflation parameter = 1.1) [50].

### 4.7. Identification of New Virophage Taxa, Delineation Criteria, and Classification Approach

To enable the identification and classification of new virophage genomes, an HMM and a BLAST database were prepared alongside adjusted cutoffs for each model and/or clade (https://github.com/simroux/ICTV_VirophageSG, accessed on 15 November 2022). First, predicted proteins from the sequence to classify are searched for the virophage morphogenesis genes using hmmsearch and the new HMM profiles (see above “Genome annotation and de novo protein clustering”). A minimum score of 50 on at least one of the MCP profiles is required to automatically classify the input sequence in the *Maveriviricetes* class. A minimum score of 40 is used for the detection of the three other virophage genes (ATPase, PRO, Penton). Next, the MCP sequences are compared to MCPs from complete and near-complete genomes robustly classified in one of the seven new groups based on the combined phylogeny and genome clustering approaches using BlastP. New sequences are affiliated with one of the seven new groups based on the best BLAST hit, with the minimum bit score cutoff determined based on the highest score obtained for sequences outside the group (Appendix A). Finally, a previously published PLV-associated HMM profile [12] was also searched using the same hmmsearch approach and a minimum score cutoff of 50 to identify PLV-like sequences. This approach was applied to the virophage dataset studied here as well as two external datasets: “Gossenköllesee” and “Adintovirus” (see above).

### 4.8. Visualization

Boxplots and bar charts were created using the ggplot2 v3.3.6 package [59] in R v4.2.1 [60], with data processed using the dplyr v1.0.10 [61] and tidyr v1.2.1 [62] packages. Annotated phylogenetic trees were generated with the packages ggtree v3.4.2 [63,64,65,66] and ggtreeExtra v1.6.1 [67]. Coloring was based on palettes from ggpubfigs [68]. Genome maps were drawn with EasyFig v2.2.3 [69] based on genome annotation from NCBI GenBank, except for PE_004, which was newly annotated here.

## Figures and Tables

**Figure 1 biomolecules-13-00204-f001:**
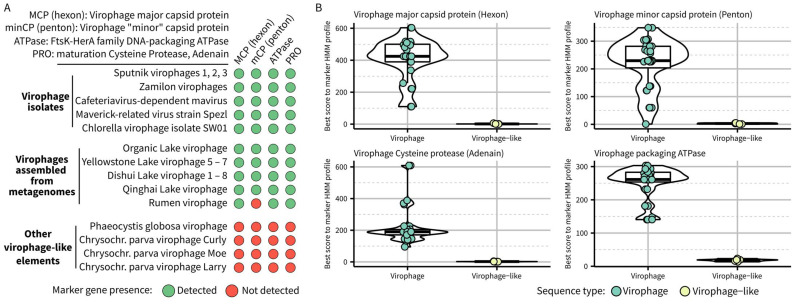
Detection of virophage marker genes across sequences currently classified in the *Lavidaviridae* family on GenBank. Marker genes were detected based on an hmmsearch performed between predicted proteins from the *Lavidaviridae* genomes and HMM profiles previously built for the four conserved genes in the virophage morphogenesis gene module [12]. Panel (**A**) displays the pattern of conserved gene detection for virophage isolates, virophage genomes assembled from metagenomes, and virophage-like elements currently classified as *Lavidaviridae* on GenBank but not encoding any of the virophage conserved genes. Panel (**B**) displays the distribution of the hmmsearch score for each marker for both the virophages (isolates or assembled from metagenomes) and the virophage-like elements. Accession numbers for the different genomes used here are indicated in Appendix A.

**Figure 2 biomolecules-13-00204-f002:**
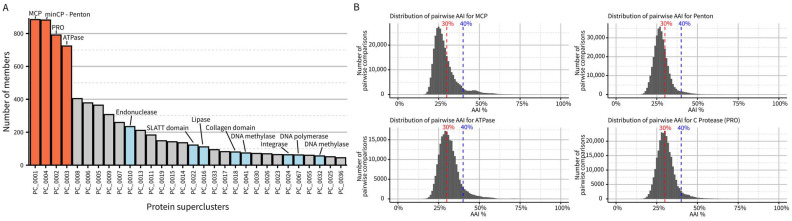
Genetic diversity in virophage genomes. (**A**) Size distribution and functional annotation of the 30 largest protein clusters obtained from the virophage sequences. The four conserved genes are colored in orange, while other protein clusters functionally annotated are colored in blue. (**B**) Distribution of pairwise amino-acid identity (AAI) across all virophage sequences for the four conserved morphogenesis genes. Two standard AAI cutoffs (30% and 40% AAI) are highlighted in red and blue, respectively. MCP: Major Capsid Protein (hexon). Penton: minor capsid protein.

**Figure 3 biomolecules-13-00204-f003:**
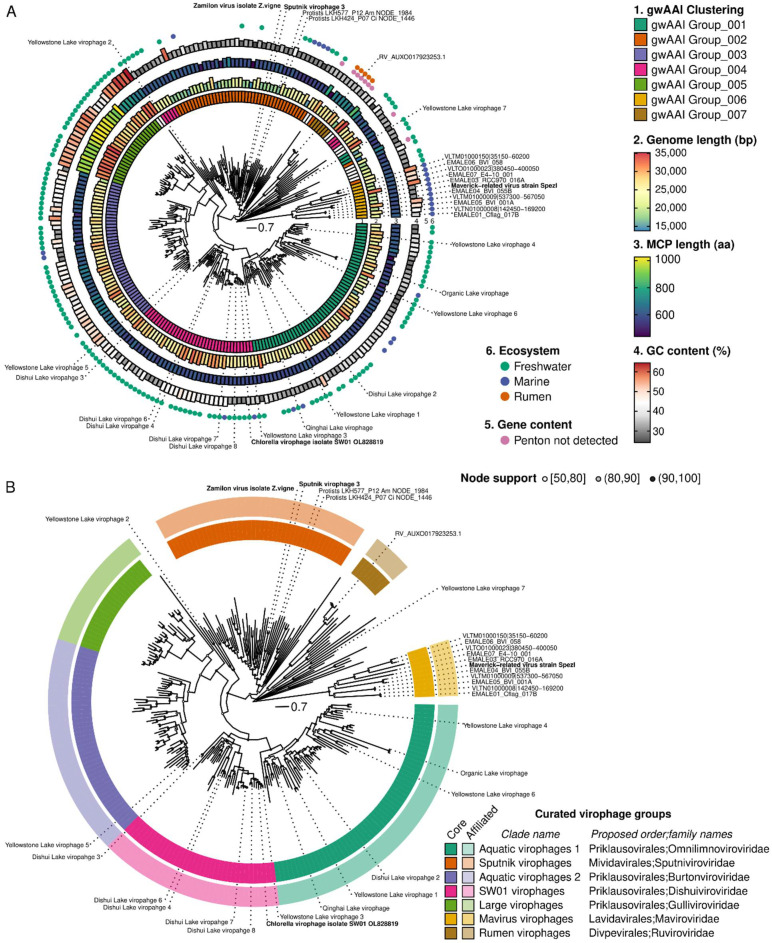
Definition of new virophage clades based on MCP phylogeny and genome features. For both panels, an MCP phylogeny is presented based on complete and near-complete virophage genomes. For panel (**A**), genomes features are displayed including from the inner to the outer ring: (1) groups based on genome-wide amino-acid identity (gwAAI) considering the seven largest groups, (2) genome length, (3) Major Capsid Protein (MCP) length, (4) average GC content, (5) detection of the typical virophage penton protein based on HMM profiles, and (6) ecosystem from which the genome was obtained. In panel (**B**), the same tree is decorated with the classification of virophage genomes in curated groups. The “core” sequences (solid colors, inner ring) of each group represent the members of the monophyletic clade assigned to this group based on the gwAAI clustering (see panel (**A**)). The “affiliated” ring displays the results of a “best BLAST hit” affiliation of all sequences in the tree (ignoring self-hits) in light colors (outer ring).

**Figure 4 biomolecules-13-00204-f004:**
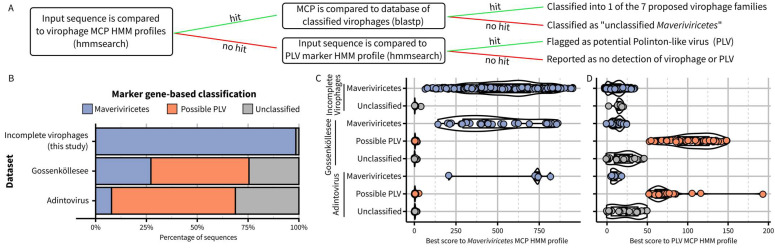
Detection and classification of virophages from published datasets. (**A**) Schematic overview of the hmmsearch- and BLAST-based classification approach used to identify and assign virophages to the proposed families. For panels (**B**–**D**), three datasets were processed with the approach described in panel (**A**). The first includes non-redundant previously published virophage sequences gathered as part of this study and not included in the complete and near-complete genome set (n = 591; “Incomplete virophages”). The second includes contigs identified as virophages and Polinton-like viruses assembled from Gossenköllesee metagenomes and published in [19] (n = 114; “Gossenköllesee”). Finally, the third dataset included adintoviruses published in several previous studies [23,32,33] (n = 64; “Adintovirus”). In panel (**B**), the percentage of sequences assigned as *Maveriviricetes* (based on MCP HMM) or with a hit to the PLV HMM profile is indicated. The score distribution of score for the hits to the *Maveriviricetes* HMM or PLV HMM is displayed in panels (**C**,**D**), respectively, for each dataset and each group of sequence.

## Data Availability

The list of all virophage genomes used in this study is provided in Appendix A, along with the associated publications and JGI proposal DOIs, when available. The corresponding sequences, along with the reference MCP alignment and phylogeny, are provided at https://portal.nersc.gov/cfs/m342/virophages/(accessed on 5 January 2023). The database and script used for the virophage sequence assignment are provided at https://github.com/simroux/ICTV_VirophageSG (accessed on 5 January 2023).

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
