# Peer review of "Updated Virophage Taxonomy and Distinction from Polinton-like Viruses"

_biomolecules, 2023, doi:10.3390/biom13020204_

Round 1

Reviewer 1 Report

Virophages are double-stranded DNA viruses that parasitize the replication cycle of a giant virus infecting a protist to replicate. They are classified within the family Lavidaviridae according the ICTV report in 2020, with 2 principal genera including genus Sputnikvirus and genus Mavirus. However, the group of virophages is expanding thanks to metagenomic studies based on different environmental and biological datasets, but also to virus isolation performed on protist culture systems. Therefore, the ICTV classification established in 2020, based on Krupovic et al. proposition in 2016, needs to be updated in order to accommodate the standing diversity of the virophage members.

In this respect, the study of Roux et al. propose the creation of one class-level group named Maveriviricetes that encompasses 7 new families of virophages. They propose to update the demarcation criteria for virophages that, namely, helps to distinguish them from polinton-like viruses. The authors also propose a sequence similarity approach in order to assign the newly virophage genomes to these clades.

The definition of virophages has been controversial and the term in this field are still confusing. The manuscript of Roux et al. is well written, easy to follow, the approach as well as the results will help to delineate these small biological entities. I strongly support their proposition and I am happy to accept this manuscript.  

Minor points:

Line 40: La Scola et al. instead of Scola, correct throughout the manuscript

Line 424 – 445: Adjust the text size in this paragraph

Author Response

We thank the reviewer for the positive feedback, and for noticing several issues with our original submission, which have now been corrected:

Line 40: La Scola et al. instead of Scola, correct throughout the manuscript
We thank the reviewer for noticing this mistake, and have now corrected these references to “La Scola et al.”

Line 424 – 445: Adjust the text size in this paragraph
We thank the reviewer for flagging this issue, and will work with the copy editor to ensure proper formatting of this section in the final version of the manuscript.

Reviewer 2 Report

This paper just satisfies the urgent needs for the classification and taxonomy of virophages, and the authors did an excellent job, especially establishing of the database and automatic classifier. Such a dynamic, growing and ready-to-use system/tool will be of great benefit to the field. Meanwhile, the paper is well written and eminently readable. I am looking forward to the demarcation criteria for taxonomic classification of virophages on the genus level by the authors in the near future.

Only few minors:

1. “1,871 sequences were clustered into 853 distinct viral operational taxonomic units (vOTUs)” (lines 170-171), while “All post-trimming virophage sequences (n=1,869) were next clustered into vOTUs” (line 562) and “The final dataset of 848 vOTU representative……” (line 568). Please check whether the discrepancy between 1871/853 and 1869/848 results from clerical errors.

2. In Figure S7, the authors said “the adintovirus similar to Sputnik virophages (bottom)”. However, according to Figure S6, it (Drosophila associated adintovirus) was clustered with mavirus. Did I misunderstand something? 

Author Response

We thank the reviewer for the positive feedback, and addressed the issues noted as follows:

1. “1,871 sequences were clustered into 853 distinct viral operational taxonomic units (vOTUs)” (lines 170-171), while “All post-trimming virophage sequences (n=1,869) were next clustered into vOTUs” (line 562) and “The final dataset of 848 vOTU representative……” (line 568). Please check whether the discrepancy between 1871/853 and 1869/848 results from clerical errors.

We apologize for the mistake, the numbers on l. 170-171 should have been “1869/848”. We have now corrected as follows l. 170: “The resulting 1,869 sequences were clustered into 848 distinct viral operational taxonomic units (vOTUs)”

2. In Figure S7, the authors said “the adintovirus similar to Sputnik virophages (bottom)”. However, according to Figure S6, it (Drosophila associated adintovirus) was clustered with mavirus. Did I misunderstand something?

We thank the reviewer for noticing this inaccuracy. Based on Fig. S6, and the results from our blast-based classifier, this virophage branches outside of any of the groups we define in this study, and thus is not assigned to a specific taxon within the Maveriviricetes class. We have modified the text accordingly l. 650: “the adintovirus identified as a virophage (Maveriviricetes) but not assigned to a specific clade”